# Choline Supplementation in Cystic Fibrosis—The Metabolic and Clinical Impact

**DOI:** 10.3390/nu11030656

**Published:** 2019-03-18

**Authors:** Wolfgang Bernhard, Robert Lange, Ute Graepler-Mainka, Corinna Engel, Jürgen Machann, Verena Hund, Anna Shunova, Andreas Hector, Joachim Riethmüller

**Affiliations:** 1Department of Neonatology, Children’s Hospital, Eberhard-Karls-University, 72076 Tübingen, Germany; robert.lange@student.uni-tuebingen.de (R.L.); anna.shunova@med.uni-tuebingen.de (A.S.); 2Department of General Pediatrics, Hematology and Oncology, Children’s Hospital, Eberhard-Karls-University, 72076 Tübingen, Germany; ute.graepler-mainka@med.uni-tuebingen.de (U.G.-M.); Andreas.Hector@med.uni-tuebingen.de (A.H.); wolfgangbernhard246@gmail.com (J.R.); 3Center for Pediatric Clinical Studies, Children’s Hospital, Eberhard-Karls-University, 72076 Tübingen, Germany; corinna.engel@med.uni-tuebingen.de; 4Department of Diagnostic and Interventional Radiology, Medical Faculty, Eberhard-Karls-University, 72076 Tübingen, Germany; juergen.machann@med.uni-tuebingen.de; 5University Pharmacy, Medical Faculty, Eberhard-Karls-University, 72076 Tübingen, Germany; verena.hund@med.uni-tuebingen.de

**Keywords:** cystic fibrosis, choline deficiency, choline supplementation, lung function, liver, steatosis, magnetic resonance spectroscopy, stable isotope labeling

## Abstract

Background: Choline is essential for the synthesis of liver phosphatidylcholine (PC), parenchymal maintenance, bile formation, and lipoprotein assembly to secrete triglycerides. In choline deficiency, the liver accretes choline/PC at the expense of lung tissue, thereby impairing pulmonary PC homoeostasis. In cystic fibrosis (CF), exocrine pancreas insufficiency results in impaired cleavage of bile PC and subsequent fecal choline loss. In these patients, the plasma choline concentration is low and correlates with lung function. We therefore investigated the effect of choline supplementation on plasma choline/PC concentration and metabolism, lung function, and liver fat. Methods: 10 adult male CF patients were recruited (11/2014–1/2016), and orally supplemented with 3 × 1 g choline chloride for 84 (84–91) days. Pre-/post-supplementation, patients were spiked with 3.6 mg/kg [methyl-D_9_]choline chloride to assess choline/PC metabolism. Mass spectrometry, spirometry, and hepatic nuclear resonance spectrometry served for analysis. Results: Supplementation increased plasma choline from 4.8 (4.1–6.2) µmol/L to 10.5 (8.5–15.5) µmol/L at d84 (*p* < 0.01). Whereas plasma PC concentration remained unchanged, D_9_-labeled PC was decreased (12.2 [10.5–18.3] µmol/L vs. 17.7 [15.5–22.4] µmol/L, *p* < 0.01), indicating D_9_-tracer dilution due to higher choline pools. Supplementation increased Forced Expiratory Volume in 1 second percent of predicted (ppFEV1) from 70.0 (50.9–74.8)% to 78.3 (60.1–83.9)% (*p* < 0.05), and decreased liver fat from 1.58 (0.37–8.82)% to 0.84 (0.56–1.17)% (*p* < 0.01). Plasma choline returned to baseline concentration within 60 h. Conclusions: Choline supplementation normalized plasma choline concentration and increased choline-containing PC precursor pools in adult CF patients. Improved lung function and decreased liver fat suggest that in CF correcting choline deficiency is clinically important. Choline supplementation of CF patients should be further investigated in randomized, placebo-controlled trials.

## 1. Introduction

Cystic fibrosis (CF) is a recessive autosomal disease, caused by mutations of the *Cystic Fibrosis Transmembrane Conductance Regulator* (*CFTR*) gene. The CFTR protein is expressed in airways, pancreatic and bile ducts, and other organs. Dysfunctional chloride and hydrogen carbonate transport in the lungs results in sticky airway secretions, with impaired mucociliary clearance, microbial colonization, inflammation, tissue degradation, and decline in lung function. Such a decline is linked to increased concentrations of pro-apoptotic ceramides, triggered by cleavage of the choline-containing phospholipid sphingomyelin (SPM) [1,2]. Ceramide removal via its conversion to SPM by SPM synthase requires PC/choline as a substrate, which may be impaired in the well-described choline deficiency of CF patients [3,4]. Moreover, exocrine pancreas insufficiency and bile duct dysfunction cause maldigestion and CF-related liver disease (CFRLD), a result of alterations from hepatosteatosis to biliary cirrhosis, in these patients [5,6,7,8,9]. Pancreatic enzyme substitution, improved nutrition, and antibiotic treatment, have increased the median life expectancy of these patients to ~38–49 years [10,11]. Nevertheless, pulmonary and hepatic failure are the most frequent causes of death, and chronic choline deficiency may contribute to clinical symptoms and impairment [4,12,13].

Choline is essential to all organs, including the lungs and liver as central organs of choline trafficking and homeostasis, mainly in the form of phosphatidylcholine (PC) and SPM. Additionally, choline is oxidized to betaine, an important methyl donor, e.g., for hepatic creatine synthesis [14]. Because endogenous PC and choline synthesis via hepatic phosphatidylethanolamine methylation is not sufficient for human requirements, choline is an essential nutrient. The higher choline requirement in CF is based on increased fecal loss, because the high amount of PC secreted into the duodenum via bile requires a functioning enterohepatic cycle of this component [15,16,17]. However, exocrine pancreas insufficiency in CF increases fecal PC loss and causes systemic choline deficiency. This is due to the lack of pancreatic phospholipase A_2_ (pPLaseA_2_) activity, precluding an efficient enterohepatic cycle of bile PC and, therefore, its choline moiety [4,12,18,19,20].

Choline is essential to pulmonary homeostasis, for the formation of PC of membranes, surfactant and secretion into the circulation to feed lipoprotein trafficking. Similarly, choline is essential to the liver, for PC of membranes, bile, and very low density lipoproteins (VLDL). A major fraction of the choline PC newly synthesized by the liver contributes to systemic lipoprotein trafficking [21,22,23,24], whereas in choline deficiency, a net PC/choline efflux from the lungs occurs to maintain liver function [25]. Such a decrease in pulmonary choline availability may impair this organ’s PC homeostasis. Consequently, recent data show that plasma choline and PC concentrations in CF patients correlate with lung function, suggesting that choline deficiency may be clinically relevant to these patients [13]. Spiking patients with [D_9_-methyl]choline in vivo shows rapid plasma choline turnover, and results in a rapid increase and decrease of plasma PC, suggesting rapid turnover of PC, too. Whereas mouse experiments showed that plasma PC kinetics reflects hepatic PC synthesis and secretion, plasma peak levels of D_9_-PC in humans were at ~24–33 h, with decreased PC secretion in CF patients [13,26,27].

We therefore supplemented, after power analysis, 10 adult male CF patients with 3 × 1 g/day choline chloride for 84 (84–91) days, to correct for their choline deficiency, and measured the effects on choline/PC homeostasis, lung function, and liver fat storage. Additionally, we spiked the patients with [D_9_-methyl]choline chloride prior to and 12 h after the end of supplementation to assess the effects of supplementation on choline and PC metabolism. As a primary outcome parameter, we investigated the effect of supplementation on plasma D_9_-PC concentration, 33 h after [D_9_-methyl]choline chloride administration. We further analyzed, as secondary parameters, the effects on lung function, liver and muscle fat, muscle creatine and choline stores, and plasma choline and PC metabolism. This pilot study is the first to integrate metabolic with clinical effects of choline supplementation in CF patients.

## 2. Materials and Methods 

### 2.1. Study Population 

Ten male CF patients from 18 to 45 years (y) of age (median: 23 years) were enrolled for this interventional trial. Patients were homozygous for Phe508del (*N* = 5) or compound heterozygous (Phe508del/1717-1G->A [*N* = 1], Phe508del/Q552X [*N* = 1], Phe508del/3849+10kbC->T [*N* = 1] and Phe508del/CFTRdele2,3(21kb) [*N* = 2]). CF was diagnosed at 2–23 months (*N* = 9) of age and at 10 y in one case (Phe508del/1717-1G->A). In all patients exocrine pancreas insufficiency was diagnosed, within 23 (*N* = 9) or at 118 months (*N* = 1, Phe508del/Phe508del) after CF diagnosis. All patients, including one with Phe508del/3849+10kbC->T, required high-dose pancreatic enzyme substitution on a daily basis. For inclusion parameters, primary and secondary end points, and anthropometric data, see Appendix A, Table A1 and Table A2. The trial was conducted in the Cystic Fibrosis Outpatient Clinic of the Tübingen University Children’s Hospital, Germany. The protocol was approved by the Institutional Review Board, and registered at ClinicalTrials.gov (Identifier: NCT 03312140). Data collection and management was performed by the Center for Pediatric Clinical Studies (CPCS), University Children’s Hospital, Tübingen. Screening was from 11/2014 to 1/2016. One hundred forty-five patients were assessed for eligibility, and 10 of them were enrolled (see Appendix A, Figure A1). 

### 2.2. Intervention 

Patients were included after written consent following visit 1 (V1). Based on previous data on choline supplementation in pediatric CF patients with 2 g/day (d) choline [28], an adequate intake of 550 mg in healthy adult males and an upper tolerable limit of 3500 mg choline per day for adult males [16], we supplemented the patients with 3 g/day choline chloride (=2.2 g/day choline). Patients were instructed to draw, three times per day, 4 mL out of a 14-mL bottle with 250 mg/mL choline chloride in sterile water (provided by the local university pharmacy), using a disposable 5-mL syringe, to dissolve it with juice, and to drink the fluid together with their regular meals. Compliance was controlled by telephone visits and by controlling for used and unused bottles at the intermediate visit (V5) (49 (45–51) d) and at the end of supplementation (V6) (84(84–91) d).

### 2.3. Outcome and Safety Parameters

The primary outcome variable was the effect of choline supplementation after the end of supplementation (V6–V8) on [D_9_-methyl]choline-labeled plasma phosphatidylcholine (D_9_-PC) concentration, 33 h after [D_9_-methyl]choline administration. Predefined secondary outcome variables were lung function and liver fat, together with the concentrations and kinetics of (D_9_-)choline, water-soluble (D_9_-)choline metabolites, (D_9_-)PC and (D_9_-)PC subgroups in plasma to assess synthesis and turnover (Appendix A, Table A2).

### 2.4. Sample Size Calculation

Based on the differences between the D_9_-PC concentration 33 h after intravenous [D_9_-methyl]choline administration in a previous study, comparing CF patients with healthy controls [13], an expected elevation of D_9_-PC from 1.468 to 2.217 µmol/L, standard deviation 0.74, alpha of 0.05 and a power of 80%, a number of 12 patients was calculated. This included the possibility of two drop-outs, which did not occur, so the study was terminated after 10 participants.

### 2.5. Investigation of Choline and Phosphatidylcholine (PC) Metabolism

To investigate the substitution effects on choline metabolism, patients were spiked orally with 3.6 mg/kg [D_9_-methyl]choline chloride (CDN Isotopes Inc., Pointe-Claire, Quebec, Canada), the same dosage as previously used intravenously [13,26], prior to supplementation (V2) and 12–15 h after the last choline chloride intake (V6). [D_9_-methyl]choline chloride was given as a single dosage, after overnight fasting, of a 50 mg/mL solution in sterile water (local university pharmacy) dissolved in 250 mL apple spritzer, together with a butter pretzel. 

### 2.6. Blood Collection and Analytical Procedures

EDTA blood (2.7 mL) was taken by venous puncture 5 min before, and at 1 h, 2 h 3 h, 6 h, 9 h, 24 h, 33 h, and 48 h after [D_9_-methyl]choline ingestion at V2–4 and V6–8, and at V5 (intermediate visit). Blood was immediately centrifuged at 1000× *g* for 10 min, and plasma and erythrocyte pellets immediately frozen at −80 °C until analysis. For analysis, 100 µL plasma was spiked with internal standards (D_4_-choline chloride, diarachinoyl-PC [PC20:0/20:0]), and extracted with chloroform and methanol [29]. Analysis was performed with electrospray ionization tandem mass spectrometry (ESI-MS/MS) as described before [21,26,27]. The chloroform phase was analyzed for unlabeled and deuterated choline-containing PC and SPM. The methanol:water phase was analyzed for unlabeled and deuterated choline, and its water-soluble metabolites betaine, dimethylglycine, and trimethylamine oxide (TMAO).

### 2.7. Analysis of Choline, Choline Metabolites, and Phospholipids 

Chloroform was of HPLC grade and from Baker (Deventer, The Netherlands). Methanol, water, and ammonium hydroxide were of analytical grade and from Fluka Analytical/Sigma-Aldrich (Munich, Germany). Phospholipid standards were from Avanti Polar Lipids (Alabaster, AL, USA) or from Sigma-Aldrich (Munich, Germany), and purity checked by tandem mass spectrometry (see below). [1,1,2,2-D_4_]choline chloride was purchased from Dr. Ehrenstorfer GmbH (Augsburg, Germany). All other chemicals were of analytical grade and from various commercial sources.

Target analytes were quantified by ESI-MS/MS, using the specific reaction monitoring (SRM) mode as described before [13,21]. Briefly, the equipment comprised a TSQ Quantum Discovery MAX triple quadrupole mass spectrometer, a Finnigan Surveyor Autosampler Plus, and a Finnigan Surveyor MS Pump Plus (Thermo Fisher Scientific, Dreieich, Germany). Choline, [D_9_-methyl]choline (D_9_-choline), betaine, [D_9_-methyl]betaine (D_9_-betaine), dimethylglycine, trimethylamine oxide (TMAO), and the internal standard (D_4_-choline) were analyzed using a ZORBAX HILIC Plus column (2.1 × 100 mm inner diameter; 3.5 µm particle size; Agilent Technologies, Böblingen, Germany) at 40 °C, a mobile phase of water:acetonitrile:formic acid (48:48:4, *v/v*) at 0.6 mL/min for 4 min, and at positive ionization in the specific reaction monitoring (SRM) [30]. Deuterated and native PC and SPM were separated on a Polaris Si-A (2 × 150 mm i.d.; 2 µm; Agilent Technologies), with chloroform:methanol:300 mM ammonium acetate (60:38:2%, *v/v*) as the mobile phase, and phosphocholine as a diagnostic fragment for PC and SPM quantification as described before [31,32].

### 2.8. Clinical Parameters

Lung function was determined using a model Jaeger Care Fusion Masterscreen body plethysmograph (Vyaire Medical, Höchberg, Germany). Forced volume vital capacity (FVC), percent of predicted forced expiratory volume in 1 s relative to FVC (ppFEV1), and forced expiratory flow at 25–75% of the pulmonary volume (FEF 25–75) were measured. A Magnetom Sonata, 1.5T (Siemens Healthineers, Erlangen, Germany) was used for magnetic resonance spectroscopy, to assess the concentrations of fat in liver and muscle tissue (anterior tibial and soleus muscle), and of muscle creatine and choline. Creatine was analyzed because its hepatic synthesis requires activated methyl groups that can originate from choline via betaine, methionine and S-adenosylmethionine [14,33].

### 2.9. Statistics

Data were controlled for normal distribution and provided as medians and interquartile ranges if not indicated otherwise. Statistical analysis was performed using Instat^®^, version 3.10 (GraphPad, La Jolla, CA, USA), using non-parametric testing. Pre-post comparisons of individual variables were analyzed with Wilcoxon matched-pairs signed-ranks test, and multiple group comparisons (V2, V5, V6) with the Kruskal‒Wallis Test. Kinetic data were analyzed with Spearman rank correlation. *P*-values are reported and *p* < 0.05 was considered significant.

## 3. Results

### 3.1. Effects of Choline Supplementation on Plasma Choline and Its Water-Soluble Metabolites

All CF patients completed the study. Heart rate, blood pressure, hemoglobin oxygenation, weight, and body mass index were in the normal range and did not change during the study. Triglycerides, cholesterol, and routine parameters of liver/kidney function and inflammation did not change either (Table A3). 

Median fasting plasma choline prior to choline supplementation and 5 min prior to [D_9_-methyl]choline labeling (V2) was 4.8(4.1–6.2) µmol/L. During supplementation at V5 (d49 (45–51)), 2–3 h after breakfast together with 1g choline chloride, plasma choline was 14.7 (11.8–16.2) µmol/L (*p* < 0.01). At V6 (d84 (84–98)), 12–15 h after the last supplementation dose, plasma choline was 10.5 (8.5–15.5) µmol/L (*p* < 0.01) (Figure 1A). Similarly, betaine, the major oxidation product of choline, was increased after choline supplementation (Figure 1B). Dimethylglycine (DMG), the demethylation product of betaine, was unchanged, whereas trimethylamine oxide (TMAO), from bacterial choline catabolism, followed by hepatic oxidation, increased in response to choline supplementation (Figure 1C,D). 

After the end of supplementation, choline, betaine and TMAO values returned to baseline within 60 h (Figure 2A–C). According to the rapid choline turnover and absorption into tissues, plasma [D_9_-methyl]choline concentration and D_9_-enrichment of choline were maximal 1 h after tracer ingestion, but rapidly cleared from the circulation (Figure 3A,B). The concentration of D_9_-betaine similarly increased (*p* < 0.001), indicating that choline supplementation stimulates its own degradation (Figure 3C). However, D_9_-enrichment of betaine was decreased, indicating that increased synthesis of D_9_-betaine after a single dose of [D_9_-methyl]choline was lower than the increase of unlabeled betaine pools after 84 days of choline supplementation (Figure 3D). This is consistent with the slower turnover of betaine compared to that of choline (Figure 2A,B, [13,26,27]). 

### 3.2. Effects of Choline Supplementation on Lung Function and Liver Triglycerides

Forced Expiratory Volume in 1 second percent of predicted (ppFEV1) was improved after choline supplementation (Figure 4), from 70.0 (50.9–74.8)% at V2 to 78.3 (60.1–83.9)% at V6, which is an absolute improvement of 8.3 (3.6–9.2)% (*p* = 0.0137). At V5 (d49 (45–51)) there was no improvement seen yet (69.9 [60.1–82.5]%). Similarly, forced vital capacity (FVC) and forced expiratory flow at 25–75% of the pulmonary volume (FEF25-75) increased, with median improvements of 5.4 (−0.3–8.2)% and 7.6 (−0.8–8.4)%, respectively, at V6. Patient no. 5, whose ppFEV1 was 86.9% at V6 compared to 90.8% at V2, apparently due to an airway infection, was included as a patient we intended to treat. 

Additionally, choline supplementation decreased liver fat from 1.6 (0.8–3.4)% (range: 0.4–8.8%) to 0.8 (0.6–1.2)% (range: 0.4–2.1%) (*p* = 0.0039), with improvements the larger the higher initial liver fat was (Figure 5A) and the highest initial value (8.8%) from the patient with a F508del/Q552X mutation. This effect was specific to the liver, because there was no effect of choline supplementation on muscle fat concentration (Figure 5B), whereas increased muscle choline and creatine, using activated (betaine-derived) methyl groups for hepatic synthesis [14] were significantly increased (Figure 5C,D). There was no significant difference between effects on CF patients homozygous for F508del and compound heterozygous patients.

### 3.3. Effects of Choline Supplementation on Plasma Phosphatidylcholine Metabolism

We further investigated whether choline supplementation increased plasma PC concentration and influenced its metabolism in these patients. Endogenous plasma PC concentration prior to choline supplementation was 1.00 (0.85–1.19) mmol/L, and was not altered after 49 (45–51) d or 84 (84–91) d choline supplementation (1.05 [0.96–1.23] and 1.01 [0.92–1.11] mmol/L, respectively). Similarly, Figure 6A shows that the molecular composition of plasma PC was identical prior to (white panels) and after (black panels) choline supplementation, with PC containing an oleic (C18:1-PC), linoleic (C18:2-PC) or arachidonic (C20:4-PC) acid residue dominating over PC containing docosahexaenoic acid (C22:6-PC).

At 33 h after [D_9_-methyl]choline administration, the plasma concentration of D_9_-PC (primary outcome parameter) was 17.7 (15.5–22.4) µmol/L before, but only 12.2 (10.5–18.3) µmol/L after choline supplementation (*p* < 0.05), reflecting higher D_9_-tracer dilution in response to choline supplementation (see Discussion).

De novo PC synthesis, reflected by D_9_-PC [13,27], resulted in preferential synthesis of C18:1-D_9_-PC and C18:2-D_9_-PC (Figure 6A, grey panels). By contrast, PC synthesis via the methylation of phosphatidylethanolamine, resulting in D_3_-PC by using (D_9_-)betaine for the synthesis of D_3_-methionine as a methyl donor [13,27] (Figure 6A, dotted panels), was preferential for the synthesis of PC containing an arachidonic (C20:4-D_3_-PC) and docosahexaenoic acid residue (C22:6-D_3_-PC). 

Although D_9_-choline enrichment of PC was decreased after choline supplementation, due to refilled choline-containing precursor pools as outlined above, plasma kinetics were identical, with maximum values at 24 h (Figure 6B). D_3_-enrichment of PC continuously increased, with identical enrichment (Figure 6B) in spite of decreased deuterium enrichment of the D_3_-methyl donor D_9_-betaine (Figure 3D). Finally, choline supplementation had no effect on the molecular specificity of PC synthesis pathways (Figure 6C), indicating that choline supplementation had no effect on the homeostasis of fatty acid or diglyceride precursors available for PC synthesis in these patients.

## 4. Discussion

Choline is essential to all cells, for membrane PC and SPM synthesis, methyl donation via betaine to synthesize methionine from homocysteine for S-adenosylmethionine (SAM) regeneration, and for other processes like osmoregulation [15]. SAM is required for hepatic PC synthesis via phosphatidylethanolamine (PE) methylation, a pathway preferential for the synthesis of C20:4-PC and C22:6-PC, and important to the assembly of very low density lipoproteins and triglyceride export from the liver [13,27]. SAM/methionine/homocysteine imbalance was related to choline deficiency in CF patients before [28]. However, the possible impact of choline deficiency on systemic PC metabolism, and on the function of organs like the liver and lungs in CF, has only recently been described [13].

Choline metabolism is characterized by rapid plasma turnover, high turnover of liver PC via the enterohepatic cycle of bile, and by pulmonary PC secretion into the circulation via transfer to apolipoprotein A1 [19,21,22,23,24,34]. In choline deficiency, choline is drained from the lungs to meet the requirements of the liver [25]. These physiological conditions impact on CF patients: exocrine pancreatic insufficiency and low duodenal pH result in decreased pPLaseA2 activity, impairing the cleavage of bile PC to lyso-PC and, therefore, the reabsorption and salvage of its choline moiety. The resulting choline deficiency may impact on the liver, and pulmonary PC/choline drainage to feed the liver may compromise epithelial integrity, repair and homeostasis of the chronically inflamed CF lung [4,12,13]

The intention of this trial was to increase the low plasma concentration of choline and PC in CF patients, as the cellular uptake and availability of choline for cellular PC synthesis is proportional to its plasma concentration [13,35,36,37]. [D_9_-methyl]choline served to assess the supplementation effect on plasma choline and hepatic PC secretion and kinetics [13,26,27]. Choline supplementation increased plasma choline from ~5 mmol/L to normal values of 10–15 µmol/L, suggesting refilled choline pools in CF patients. Whereas choline supplementation did not increase plasma PC concentration, it decreased D_9_-choline enrichment of PC and total D_9_-PC concentration after [D_9_-methyl]choline intake. This suggests stronger dilution of the D_9_-tracer, and supports the concept of increased tissue concentrations of PC precursors (choline, phosphocholine, CDP-choline [37]) in response to supplementation. Hence, 3 × 1 g/day choline chloride, equaling 2.2 g choline, the 4-fold adequate intake of healthy adult men [16], effectively ameliorated choline deficiency in CF patients. This is in line with the improvement of the methionine/homocysteine status in response to a similar dosage of choline in pediatric CF patients [28].

Although being secondary outcome parameters in this study, improved lung function and decreased liver fat indicate the clinical relevance of correcting choline deficiency in CF patients. Improved ppFEV1 after 84 d choline supplementation is consistent with previous data showing the direct correlation of plasma choline and lung function in CF, and suggest a causal relationship rather than an association [13]. 

The mechanisms involved, however, require further investigation, but may relate to the improved choline availability of the lungs. In this organ, PC is exclusively synthesized de novo from exogenous choline uptake, proportional to its extracellular concentration [15,35,36,38]. Beyond its functions in membranes and surfactant, and its secretion into the circulation, via basolateral ATP-binding cassette transporter A1 (ABC-A1), to feed systemic lipoprotein homeostasis and the liver in choline deficiency [22,23,24,25], PC serves to remove pro-apoptotic ceramides via sphingomyelin (SPM) synthase (EC2.7.8.27) [3]. In CF lungs, such ceramides are increased via SPM cleavage by acid sphingomyelinase (ASM) (EC3.1.4.12), which causes poor lung function. Decreasing ceramide production by ASM inhibition significantly reduces inflammation and epithelial apoptosis, and improves lung function [1,2]. Increasing the backward reaction of ceramide removal by SPM synthase requires sufficient amounts of PC as a substrate. Correcting for the choline deficiency is a plausible mechanism to improve lung function in these patients, which requires further investigation. A ferret or pig CFTR knockout model, mimicking both the gastrointestinal and pulmonary pathology of human CF more closely than mouse models [39,40], may be useful to further investigate the mechanisms involved. Experiments will have to show whether impaired pancreas function results in choline deficiency as a general mechanism impacting on lung and liver function in CF, and whether choline supplementation will improve pulmonary choline, PC and ceramide homeostasis as well as chronic inflammation and loss of organ function.

Choline supplementation diminished liver fat, too. PC synthesis via de novo synthesis and methylation of phosphatidylethanolamine (PE) via PE-N-methyltransferase (PEMT) are essential for hepatic triglyceride export [19]. Whereas, de novo synthesis directly uses choline, the PEMT pathway uses S-adenosylmethionine (SAM), whose methyl groups can be derived from choline via betaine, and results in D_3_- rather than D_9_-PC [13,27,41]. Both of these were measurable prior to and after choline supplementation, suggesting the presence of both pathways in CF liver as shown before [13]. The decreased in liver triglycerides suggests that during choline supplementation increased liver triglycerides were exported. Whereas plasma PC and triglycerides were not increased, concentrations were measured after 49 d (V5) and 84 d (V6), when triglyceride export from liver was accomplished. 

Such supplementation effects on triglycerides were specific for the liver, as triglycerides were not changed in muscle tissue, whereas free choline and creatine, requiring SAM as a methyl donor for hepatic synthesis [14], were increased in muscle tissue. These data further suggest a systemic improvement of the choline status in these patients. Moreover, synthesis of D_3_-PC (see Figure 6B and [13,27]), which requires (D_3_-)SAM, proves that D_9_-choline is used as a methyl donor for D_3_-SAM synthesis from homocysteine and D_9_-betaine. Hence, choline supplementation not only increases its availability for PC synthesis but also for methyl donation via betaine. By contrast, choline supplementation did not change the fatty acid pattern in plasma PC, particularly not the fraction of C20:4-PC and C22:6-PC, their primarily transport vehicle in plasma. Hence, choline supplementation does not impact on the potential DHA/ARA imbalance in CF patients [13,42,43,44].

### Limitations

This is a pilot study on the metabolic and clinical impact of correcting for the choline deficiency in CF patients with exocrine pancreatic insufficiency, by choline supplementation based on clinical and experimental data, and on the application of pathophysiological considerations [1,2,3,13]. Whereas this concept proved reasonable, choline turnover was very rapid so that plasma choline, betaine and TMAO decreased to baseline concentrations within 60 h after the last choline dose (V6–V8). Hence, the rapid choline turnover is limiting and may require an approach superior to choline chloride. Whether other choline carriers, like glycerophosphorylcholine or PC, are superior to the plasma kinetics free choline salts [45,46], will have to be addressed in humans. Moreover, this trial was not randomized or placebo-controlled, and only blinded for the pre-post-analysis of the primary and biochemical outcome parameters (plasma D_9_-PC etc.). There was no follow-up in this trial, which will have to be included in future, and it was conducted on a rather homogenous and stable collective of adult patients with ppFEV1 > 40% in males. The reasoning here was that *PEMT* gene expression, which reduces exogenous choline requirements, is higher in pre-menopausal women compared to men, due to its stimulation by estrogens. On the other hand, single nucleotide polymorphisms of this gene are frequent, making female CF patients a more heterogenous collective [19,47,48]. Moreover, while *PEMT* gene expression by high estrogen levels does not apply to children, their choline requirements are higher due to growth requirements [15,16,18]. Finally, our study collective comprised CF patients homozygous for F508del as well as F508del compound heterozygous patients with exocrine pancreas insufficiency. In essence, our results serve as a proof of principle, but generalization of the effect of choline supplementation in CF patients awaits further investigation in clinical trials addressing the effects of choline supplementation in men, women, and children and with different CFTR genotypes. Importantly, the structural and functional impairment of the exocrine pancreas occurs rapidly after birth, suggesting that early intervention to improve choline homeostasis may be useful. Routine finding of low plasma choline in pediatric patients of our outpatient CF clinic and others before [49] support this view. While novel CFTR channel correctors significantly improve lung function in CF patients, nothing is known about their effects on systemic choline homeostasis [50,51]. Hence, choline supplementation appears to be an option for CF treatment, complementary to other innovative therapies.

## 5. Conclusions and Perspective

Choline supplementation normalizes the decreased concentration of plasma choline and of PC precursors in the liver of CF patients with exocrine pancreas insufficiency. It improves lung function and hepatosteatosis in these patients, complementary to standard care and, possibly, other therapeutic innovations. The relevance of choline as an essential nutrient for parenchymal growth and homeostasis suggests its supplementation in children [15]. The rapid return of plasma choline to baseline concentration suggests further efforts to optimize the galenics of choline and to assess the duration of clinical improvements in randomized trials.

## Figures and Tables

**Figure 1 nutrients-11-00656-f001:**
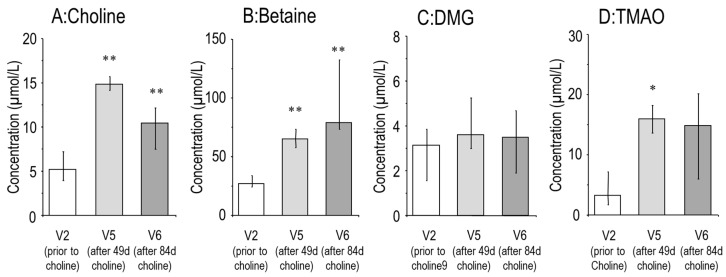
Effect of oral choline supplementation on the plasma concentration of choline (**A**) and its water-soluble metabolites betaine, dimethylglycine (DMG) and trimethylamine oxide (TMAO) (**B–D**). Bars represent values prior to (visit [V] 2), after 49 (45–51) days (V5) and after 84 (84–91) days (V6) supplementation with 3 × 1 g choline chloride. Data are medians and interquartile ranges of CF patients prior to (V2), and after 49 (45–51) days (V5) and 84 (84–91) days (V6) choline supplementation. * *p* < 0.05; ** *p* < 0.01 vs. V2.

**Figure 2 nutrients-11-00656-f002:**
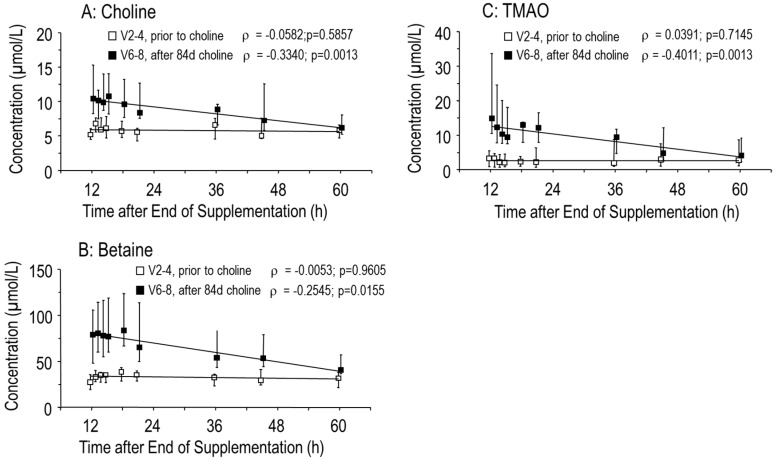
Plasma concentrations of choline (**A**) and its metabolites betaine (**B**) and trimethylamine oxide (TMAO) (**C**). Data are values of 10 adult male CF patients over 48 h prior to choline supplementation (visit 2–4 [V2–4], white symbols) and from 12 h after the end of choline supplementation (84 (84–91) days) onwards (V6–8, filled symbols). Abbreviations: *ρ*, Spearman rank correlation coefficient; *p*, significance level.

**Figure 3 nutrients-11-00656-f003:**
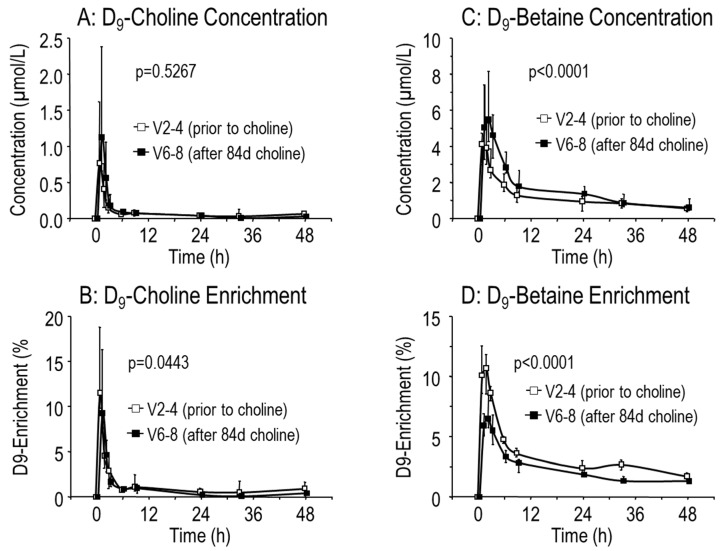
Kinetics of D_9_-Choline and D_9_-Betaine prior to (visit 2–4 [V2–4], white symbols) and from 12 h after the end of 84 (84–91) days choline supplementation (V6–8, filled symbols). CF patients (*N* = 10) were orally supplemented with 3 × 1 g choline chloride for 84 (84–91) d. Prior to (V2, open symbols) and 12 h after the end of supplementation (V6, filled symbols) patients were spiked orally with 3.6 mg/kg body weight [D_9_-methyl]choline chloride (D_9_-choline) and the kinetics of plasma D_9_-choline (**A**) and D_9_-betaine (**C**) concentration and of D_9_-enrichment of choline (**B**) and betaine (**D**) analyzed for 48 h (V2–4 and V6–8, respectively). Data are medians and interquartile ranges from −5 min to 48 h after D_9_-choline ingestion. Abbreviations: *p*, significance level.

**Figure 4 nutrients-11-00656-f004:**
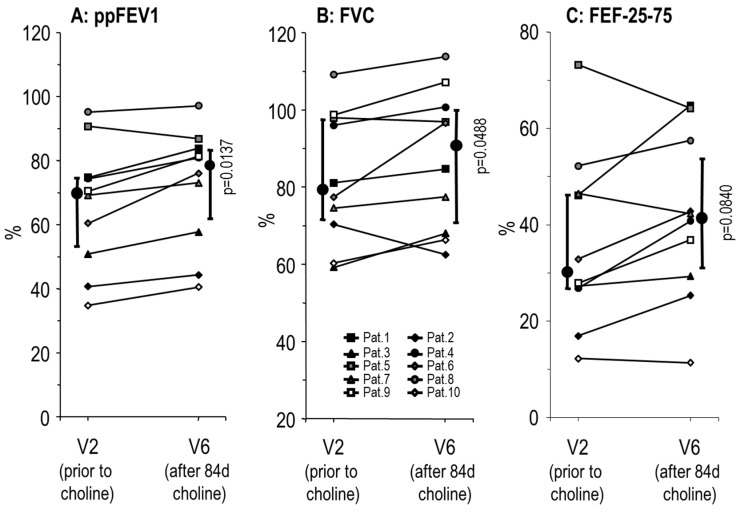
Effects of choline supplementation on lung function. Data indicate values prior to (V2) and after 84 (84–91) days (V6) supplementation with 3 × 1 g/day choline chloride. Data indicate age-corrected Relative Forced Expiratory Volume in 1 second (Tiffeneau Index; ppFEV1) (**A**), Forced Vital Capacity (FVC) (**B**) and Forced Expiratory Flow at 25–75% of the pulmonary volume (FEF25-75) (**C**). Small symbols and thin lines indicate individual values of patients, whereas large symbols and thick bars indicate medians and interquartile ranges. *p* significance level.

**Figure 5 nutrients-11-00656-f005:**
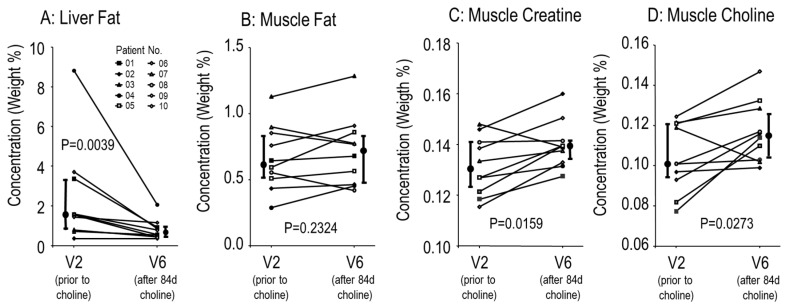
Effects of choline supplementation on liver and muscle tissue. Liver (**A**) and muscle fat (**B**), and muscle creatine (**C**) and choline (**D**) were investigated with magnetic resonance spectroscopy as shown in Materials and Methods. Small symbols and thin lines indicate individual values of patients, whereas large symbols and thick bars indicate medians and interquartile ranges. *p*, significance level.

**Figure 6 nutrients-11-00656-f006:**
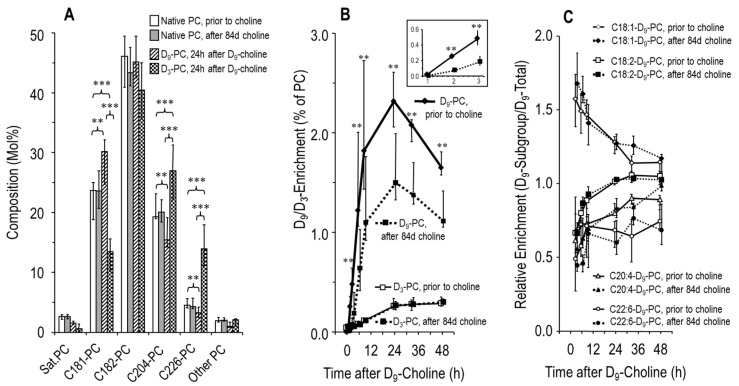
Molecular composition and kinetics of plasma phosphatidylcholine (PC) in response to choline supplementation. (**A**) PC molecular species grouped according to their content of different fatty acid residues in sn-position 2 (Sat. PC, C18:1-PC, C18:2-PC, C20:4-PC or C22:6-PC). White panels indicate the median PC composition prior to choline supplementation (visit [V] 2‒4, whereas grey panels are values after 84 d choline supplementation (V6-8). Striped panels indicate the molecular composition of PC synthesized de novo (D_9_-PC), whereas dotted panels indicate synthesis via methylation of phosphatidylethanolamine (D_3_-PC) [13,27]. (**B**) D_9_- and D_3_-enrichment of PC prior to and after choline supplementation. (**C**) The relative synthesis rates and kinetics of plasma PC sub-groups are identical prior to (visit [V] 2–4) and after (V6–8) 84 d choline supplementation. Data are medians and interquartile ranges of *N* = 10. Abbreviations: ** *p* < 0.01; *** *p* < 0.001.

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
