# Peer review of "Choline Supplementation in Cystic Fibrosis—The Metabolic and Clinical Impact"

_nutrients, 2019, doi:10.3390/nu11030656_

Reviewer 1 Report

This is a well executed and written report of a pilot study on n = 10 adult males with CF with 508/508 mutations and the results of short term choline supplementation on a number of biochemical and clinically meaningful outcomes. 

Please add information about the exocrine pancreatic status of the subjects (fecal elastase or CFA, etc).

Given that this was a report of a pilot study of only male adults, consider limited the 'strength of the conclusions'.  and agree that further study is needed.

Author Response

Reviewer 1: This is a well executed and written report of a pilot study on n = 10 adult males with CF with 508/508 mutations and the results of short term choline supplementation on a number of biochemical and clinically meaningful outcomes. 

Answer: We thankfully acknowledge the reviewer’s critical acclaim of our trial.

Reviewer 1: Please add information about the exocrine pancreatic status of the subjects (fecal elastase or CFA, etc).

Answer: The authors thankfully acknowledge the reviewer’s question about the quantification of exocrine pancreas insufficiency (EPI). However, while EPI was an  inclusion parameter, and was verified from patient files and regular intake of pancreatic enzymes (such as Kreon®), measurement of pancreas elastase was not performed during screening or after inclusion of patients, nor was it routinely done in our patients with a history of  >10y after CF or EPI diagnosis. Only in two patients, pancreas elastase in stool was verified, with values<15µg/g, in 2002 and 2006, respectively. Hence, we unfortunately cannot provide actual data on this or related parameters. Enzyme intake, necessary for weight maintenance and regular digestion, was used as a proxy for requiring exogenous pancreatic enzymes due to EPI.

When analyzing the patient files in detail, the corresponding author (W.B., project leader) became aware that the description of patients in the first submission was incorrect. Only part of the patients included by the principle investigator (J. R.) were Phe508del homozygous, as only EPI/enzyme substitution and not Phe508del homozygosity was an inclusion parameter. Surely, if not having deceased 1 ½ years ago, he would have corrected for this error in time. The corresponding author takes full responsibility for this error and has corrected for it in this revision.

We have controlled for differences between the homozygous and compound heterozygous patients for lung function improvement and liver fat reduction, without significant differences. This is outlined on page 3, para 1, line 97ff (Study population) and mentioned in Results, p. 7, last para, line 248-249.

Reviewer 1: Given that this was a report of a pilot study of only male adults, consider limited the 'strength of the conclusions'  and agree that further study is needed.

Answer: We completely agree, and we have included the limitations of the study more detailed in the discussion. See Limitations, page 11,  line 388ff.

Reviewer 2 Report

This is a review of the manuscript entitled, “Choline Supplementation in Cystic Fibrosis – Metabolic and Clinical Impact”, by Bernhard, et al. This is an interesting study that tests whether choline supplementation could affect people with CF in a clinically relevant manner.

Major comments

none

Minor comments

Did the authors evaluate whether the withdraw of choline supplementation resulted in reversal of the clinical improvement?

Have the authors considered utilizing relevant CF models that get both lung and GI disease, such as the pig (doi: 10.1126/scitranslmed.3000928) and/or ferret (doi: 10.1172/JCI43052), to better clarify the clinical outcomes and test the mechanisms (e.g. lung tissue PC levels, immune system effects, etc.) for the choline supplementation? If not, the authors might discuss this and other ways to further development of this potentially useful adjunct therapy for clinical application.

For the methods or in the figures, appendices and tables – including/clarifying the time relative to choline supplementation in addition with the Visit numbers (e.g. “V2”, etc.) – would help the reader in interpreting the results. Perhaps even as a subtitle below the figure/table (Esp say Table A2 and Appendix B), or even a simple supplementation figure table to clarify these “V” numbers and give an overall experimental design in relation to Choline supplementation.

318 – “Although” is used twice in this sentence…..

367 - Could the authors briefly discuss more the differences in sex/age in choline metabolism that was mentioned in the discussion.

380-382 – clarify this sentence – it sounds incomplete…

Author Response

Reviewer 2-Comments and Suggestions for Authors

This is a review of the manuscript entitled, “Choline Supplementation in Cystic Fibrosis – Metabolic and Clinical Impact”, by Bernhard, et al. This is an interesting study that tests whether choline supplementation could affect people with CF in a clinically relevant manner.

Answer: The authors thankfully acknowledge the appraisal of our work.

Major comments

None

Answer: Thank you

Minor comments

Reviewer 2: Did the authors evaluate whether the withdraw of choline supplementation resulted in reversal of the clinical improvement?

Answer: Thank you! At this stage of research the authors have not performed any follow-up of patients. However, this is in our agenda for future trials. A sentence was included in Discussion, Limitations, page 11, line 388f.

Reviewer 2: Have the authors considered utilizing relevant CF models that get both lung and GI disease, such as the pig (doi: 10.1126/scitranslmed.3000928) and/or ferret (doi: 10.1172/JCI43052), to better clarify the clinical outcomes and test the mechanisms (e.g. lung tissue PC levels, immune system effects, etc.) for the choline supplementation? If not, the authors might discuss this and other ways to further development of this potentially useful adjunct therapy for clinical application.

Answer: Thank you! A passage suggesting experiments to evaluate choline deficiency and the mechanisms of supplementation on lung function and liver steatosis improvements in these animal models has been included. See Discussion, page 10, last para, line 352-357.

Reviewer 2: For the methods or in the figures, appendices and tables – including/clarifying the time relative to choline supplementation in addition with the Visit numbers (e.g. “V2”, etc.) – would help the reader in interpreting the results. Perhaps even as a subtitle below the figure/table (Esp say Table A2 and Appendix B), or even a simple supplementation figure table to clarify these “V” numbers and give an overall experimental design in relation to Choline supplementation.

Answer: Thank you for this advice! We have changed the tables, figures and legends accordingly so that the time of measurement is related to choline supplementation (prior to choline/after 49d choline/after 84d choline)

Reviewer 2: 318 – “Although” is used twice in this sentence…..

Answer: Thank you! We have removed the part with the second “although”. See line 335ff.

Reviewer 2: 367 - Could the authors briefly discuss more the differences in sex/age in choline metabolism that was mentioned in the discussion.

Answer: We are thankful for this comment. We have addressed this in more detail in the Discussion, Limitations, line 390-395.

Reviewer 2: 380-382 – clarify this sentence – it sounds incomplete…

Answer: The wording of the sentence was changed accordingly. Line 412-414.